# Horizontal Transposon Transfer and Its Implications for the Ancestral Ecology of Hydrophiine Snakes

**DOI:** 10.3390/genes13020217

**Published:** 2022-01-25

**Authors:** James D. Galbraith, Alastair J. Ludington, Kate L. Sanders, Timothy G. Amos, Vicki A. Thomson, Daniel Enosi Tuipulotu, Nathan Dunstan, Richard J. Edwards, Alexander Suh, David L. Adelson

**Affiliations:** 1School of Biological Sciences, University of Adelaide, Adelaide, SA 5005, Australia; j.d.galbraith@exeter.ac.uk (J.D.G.); alastair.ludington@adelaide.edu.au (A.J.L.); kate.sanders@adelaide.edu.au (K.L.S.); vickithmsn@gmail.com (V.A.T.); 2Centre for Ecology and Conservation, University of Exeter, Penryn Campus, Cornwall TR10 9FE, UK; 3School of Biotechnology and Biomolecular Sciences, University of New South Wales, Sydney, NSW 2052, Australia; t.amos@garvan.org.au (T.G.A.); daniel.enosi@anu.edu.au (D.E.T.); 4Garvan Institute of Medical Research, Sydney, NSW 2010, Australia; 5Division of Immunity, Inflammation and Infection, The John Curtin School of Medical Research, The Australian National University, Canberra, ACT 2601, Australia; 6Venom Supplies Pty Ltd., Tanunda, SA 5252, Australia; nathan@venomsupplies.com; 7School of Biological Sciences, University of East Anglia, Norwich Research Park, Norwich NR4 7TU, UK; 8Department of Organismal Biology-Systematic Biology, Evolutionary Biology Centre, Uppsala University, SE-752 36 Uppsala, Sweden; 9South Australian Museum, Adelaide, SA 5000, Australia

**Keywords:** transposable element, comparative genomics, Serpentes

## Abstract

Transposable elements (TEs), also known as jumping genes, are sequences able to move or copy themselves within a genome. As TEs move throughout genomes they often act as a source of genetic novelty, hence understanding TE evolution within lineages may help in understanding environmental adaptation. Studies into the TE content of lineages of mammals such as bats have uncovered horizontal transposon transfer (HTT) into these lineages, with squamates often also containing the same TEs. Despite the repeated finding of HTT into squamates, little comparative research has examined the evolution of TEs within squamates. Here we examine a diverse family of Australo–Melanesian snakes (Hydrophiinae) to examine if the previously identified, order-wide pattern of variable TE content and activity holds true on a smaller scale. Hydrophiinae diverged from Asian elapids ~30 Mya and have since rapidly diversified into six amphibious, ~60 marine and ~100 terrestrial species that fill a broad range of ecological niches. We find TE diversity and expansion differs between hydrophiines and their Asian relatives and identify multiple HTTs into Hydrophiinae, including three likely transferred into the ancestral hydrophiine from fish. These HTT events provide the first tangible evidence that Hydrophiinae reached Australia from Asia via a marine route.

## 1. Introduction

Variation is the fundamental basis of all evolutionary change, and mobile genetic elements are a major source of genomic variation. A high proportion of animal and plant genome sequences is derived from transposable elements (TE) and TEs are acknowledged drivers of evolutionary change, but their impacts are poorly understood. Understanding how TEs drive evolutionary change requires studying systems that are young, species-rich and ecologically diverse. In these respects, elapid snakes present excellent opportunities for the study of TE dynamics and their contribution to adaptive changes. Elapids are a diverse group of venomous snakes found across Africa, Asia, the Americas and Australia. Following their divergence from Asian elapids ~30 Mya, the Australo–Melanesian elapids (Hydrophiinae) have rapidly diversified into more than 160 species including ~100 terrestrial snakes, ~60 fully marine sea snakes and six amphibious sea kraits [1]. Both the terrestrial and fully marine hydrophiines have adapted to a wide range of habitats and niches. Terrestrial Hydrophiinae are found across Australia, for example, the eastern brown snake (*Pseudonaja textilis*) in open habitats, the tiger snake (*Notechis scutatus*) in subtropical and temperate habitats and the inland taipan (*Oxyuranus microlepidotus*) in inland arid habitats [2]. Sea snakes are phylogenetically closer to tiger snakes than the other terrestrial Hydrophiinae, so share a common ancestor. Since transitioning to a marine habitat, many sea snakes have specialized to feed on a single prey such as fish eggs, catfish, eels or burrowing gobies, while others such as *Aipysurus laevis* are generalists [3,4] (hereafter, all mentions of these species will use the genus name only, i.e., *Aipysurus* for *A. laevis*). Sea kraits (*Laticauda*) are amphibious and have specialized to hunt various fish including eels and anguilliform-like fish at sea, while digesting prey, mating and shedding on land [5]. Since transitioning to marine environments, both sea snakes and sea kraits have been the recipients of multiple independent horizontal transposon transfer (HTT) events, with adaptive potential [6,7].

Transposable elements (TEs) are mobile genetic elements that can move or copy themselves across the genome, and account for a large portion of most vertebrate genomes [8,9]. Though often given short shrift in genome analyses, TEs are important agents of genome evolution and generate genomic diversity [10,11]. For example, the envelope gene of an endogenous retrovirus was repeatedly exapted by both mammals and viviparous lizards to function in placenta development [12]. In addition, unequal crossing over caused by CR1 retrotransposons led to the duplication, and hence diversification, of the PLA_2_ venom genes in pit vipers [13].

Transposable elements (TEs) are classified into two major classes based on their structure and replication method [14]. DNA transposons (Class II) mostly proliferate through a “cut and paste” method, possess terminal inverted repeats and are further split based on the transposase sequence used in replication. Retrotransposons (Class I) are split into LTR retrotransposons and non-LTR retrotransposons, which proliferate through “copy and paste” methods. Both subclasses of retrotransposons are split into numerous superfamilies based on both coding and structural features [15,16,17]. Within the diverse lineages of land vertebrates, the evolution of TEs is well described in eutherian mammals and birds. The total repetitive content of both bird and mammal genomes is consistently at 7–10% and 30–50%, respectively. Similarly, most lineages of both birds and eutherian mammals are dominated by a single superfamily of non-LTR retrotransposons (CR1s and L1s, respectively) and a single superfamily of LTR retrotransposons (endogenous retroviruses in both) [8,18]. Some lineages of birds and mammals contain horizontally transferred retrotransposons that have been variably successful (AviRTE and RTE-BovB, respectively) [19,20].

In stark contrast to mammals and birds, squamates have highly variable mobilomes, both in terms of the diversity of their TE superfamilies and the level of activity of said superfamilies within each genome [21]. While these broad comparisons have found significant variation in TEs between distant squamate lineages, none have examined how TEs have evolved within a single family of squamates. The one in-depth study into the mobilome of snakes found that the Burmese python genome is approximately ~21% TE and appears to have low TE expansion, while that of a pit viper is ~45% TE, due to the expansion of numerous TE superfamilies and microsatellites since their divergence ~90 Mya [22,23]. It thus is unclear whether similar expansions have occurred within other lineages of venomous snakes. Here, we examine the TE landscape of the family Hydrophiinae, and in doing so discover horizontal transfer events into the ancestral hydrophiine, sea kraits, sea snakes and tiger snakes.

## 2. Materials and Methods

### 2.1. Sample Collection, Library Construction and Sequencing of Terrestrial Elapids

Terrestrial elapids were provided by Venom Supplies Pty Ltd. Collection of animal tissue was approved by the Australian National University Animal Experimentation Ethics Committee (approval number R.CG.14.08). The mainland tiger snake (*Notechis scutatus*) was caught in southeast South Australia in 2004, just north of Mt Gambier. The eastern brown snake (*Pseudonaja textilis*) was bred at Venom Supplies from a female snake that had been gravid when caught locally in the Barossa. High-molecular-weight genomic DNA (gDNA) was extracted from the blood (*N. scutatus*) or tail clip (*P. textilis*) using a genomic-tip 100/G kit (Qiagen, Hilden, Germany). This was performed with proteinase K (NEB, Ipswich, MA, USA) to digest the sample and treated with RNase A (Astral Scientific, Taren Point, NSW, Australia) to remove RNA, as per the manufacturer’s instructions. Isolated genomic DNA was further purified using AMPure XP beads (Beckman Coulter, Brea, CA, USA) to eliminate sequencing inhibitors. DNA quantity was assessed using the Quant-iT PicoGreen dsDNA kit (Thermo Fisher Scientific, Waltham, MA, USA), DNA purity was calculated using a Nanodrop spectrophotometer (Thermo Fisher Scientific) and molecular integrity assessed by pulse-field gel electrophoresis. For each snake, a 10× genomics linked-read library was prepared using the Chromium™ Genome Reagent Kit (v2 Chemistry) for 2 × 150 bp paired-end sequencing on the HiSeq × Ten (llumina) platform.

### 2.2. Genome Assembly of Terrestrial Elapids

For each snake, raw 10× linked-reads were assembled using Supernova v2.0.0 (10× Genomics) using default settings. Additional cleanup of the pseudohap2 output was performed as described in [24] to generate a non-redundant primary and alternative assembly for each species. In each case, the primary assembly contains the longest of each pair of phased haplotigs. Purely homozygous haplotigs were only included in the primary assembly. Genome assemblies were annotated by NCBI.

### 2.3. Genome Completeness and Summary Statistics

Genome completeness was estimated for each snake genome using BUSCO v5.0.0 [25] and MetaEuk v732bcc4b91a08e69950ce0e25976f47c3bb6b89d [26] was used for gene prediction (BLAST+ v2.11.0 [27], SEPP v4.3.10 [28] and Hmmer (Hmmer, RRID:SCR_005305) v3.3 [29]), against the sauropsida_odb10 dataset (*n* = 7840). BUSCO results were compiled and the assembly stats generated using BUSCOMP [24].

### 2.4. Ab Initio TE Annotation of the Elapid Genomes

We used RepeatModeler2 [30] to perform ab initio TE annotation of the genome assemblies of four hydrophiines (*Aipysurus laevis*, *Notechis scutatus*, *Pseudonaja textilis* and *Laticauda colubrina*) and two Asian elapids (*Naja naja* and *Ophiophagus hannah*). We manually curated the subfamilies of TEs identified by RepeatModeler (rm-families) to ensure they encompassed the full TE, were properly classified and that each species’ library was non-redundant.

We first purged redundant rm-families from each species’ library based on pairwise identity to and coverage by other rm-families within the library. Using BLASTN 2.7.1+ [31] we calculated the similarity between all rm-families. Any rm-family with over 75% of its length aligning to a larger rm-family at 90% pairwise identity or higher was removed from the library. We then searched for each non-redundant rm-family within their source genome with BLASTN 2.7.1+ (-task dc-megablast) and selected the best 30 hits based on bitscore. In order to ensure we could retrieve full length TE insertions, we extended the flanks of each hit by 4000 bp. Using BLASTN 2.7.1+ (-task dc-megablast) we pairwise aligned each of the 30 extended sequences to others, trimming trailing portions of flanks that did not align to flanks of the other 29 sequences. Following this, we constructed a multiple sequence alignment (MSA) of the 30 trimmed sequences with MAFFT v7.453 [32] (localpair). Finally, we trimmed each MSA at the TE target site duplications (TSDs) and constructed a consensus from the multiple sequence alignments using Geneious Prime 2021.1.1 (www.geneious.com) (accessed on 6 January 2022), which we hereafter refer to as a mc-subfamily (manually curated subfamily).

To classify the mc-subfamilies we searched for intact protein domains in the consensus sequences using RPSBLAST 2.7.1+ [33] and the CDD library [34] and identified homology to previously described TEs in Repbase using CENSOR online [35]. Using this data in conjunction with the classification set out in Wicker (2007) [14], we classified previously unclassified mc-subfamilies where possible and corrected the classification of mc-subfamilies where necessary. Where possible we used the criteria of Feschotte and Pritham (2007) [16] to identify unclassified DNA transposons using TSDs and terminal inverted repeats. Finally, we removed any genes from the mc-subfamily libraries based on searches using online NCBI BLASTN and BLASTX searches against the nt/nr and UniProt libraries, respectively [36,37]. Any mc-subfamilies unable to be classified were labeled as “Unknown”.

### 2.5. TE Annotation of the Elapid Genomes

We constructed a custom library for TE annotation of the elapid genome assemblies by combining the mc-subfamilies from the six assemblies with previously described lepidosaur TEs identified using Repeat Masker’s “queryRepeatDatabase.pl” utility. Using Repeat Masker, we generated repeat annotations of all six elapid genome assemblies.

### 2.6. Estimating Ancestral TE Similarity

To estimate the sequence conservation of ancestral TEs, and hence categorize recently expanding TEs as either ancestral or horizontally transferred, we identified orthologous TE insertions and their flanks present in both the *Notechis* and *Naja* genome assemblies. From the *Notechis* repeat annotation, we took a random sample of 5000 TEs over 500 bp in length and extended each flank by 1000 bp. Using BLASTN (-task dc-megablast) we searched for the TEs and their flanks in the *Naja* assembly and selected all hits containing at least 250 bp of both the TE and the flank. Sequences with more than one hit containing flanks were treated as potential segmental duplications. We also removed any potential segmental duplications from the results. We then used the orthologous sequences to estimate the expected range in similarity between TEs present in the most recent common ancestor of Australian and Asian elapids. Based on this information, TEs with 95% or higher pairwise identity to the mc-subfamily used to identify them were treated as likely inserted in hydrophiine genomes since their divergence from Asian elapids. In addition, mc-subfamilies, which we had identified as recently expanding in hydrophiines but were not found at 80% or higher pairwise identity in other serpentine genomes, were identified as candidates for horizontal transfer.

### 2.7. Identifying Recent TE Expansions

In each of the four hydrophiines, using the Repeat Masker output we identified mc-subfamilies comprising at least 100 kb total having 95% or higher pairwise identity to the mc-subfamily. We treated these mc-subfamilies as having expanded since Hydrophiinae’s divergence from Asian elapids. We reduced any redundancy between recently expanding mc-subfamilies by clustering using CD-HIT-EST (-c 0.95 -n 6 -d 50) [38]. Using BWA [39], we mapped raw transcriptome reads from eye tissue of each of the hydrophiines [40] back to these mc-subfamilies. Retrotransposons with RNA-seq reads mapping across their whole length and DNA transposons with RNA-seq reads mapping to their coding regions were treated as expressed and therefore currently expanding.

### 2.8. Continued Expansion or Horizontal Transfer

Using BLASTN (-task dc-megablast), we searched for homologs of recently expanding mc-subfamilies in a range of snake genomes including Asian elapids, colubrids, vipers and a python. We classified mc-subfamilies having copies of 80% or higher pairwise identity to the query sequence in other snakes as ancestral. All hydrophiine mc-subfamilies we were unable to find in other snakes were treated as candidates for horizontal transfer. Using BLASTN (-task dc-megablast), we searched for the horizontal transfer candidates in over 2300 additional metazoan genomes from Genbank (Appendix A). We classified all mc-subfamilies present in non-serpentine genomes at 80% or higher pairwise identity and absent from other serpentine genomes at 80% or higher pairwise identity as horizontally transferred into hydrophiines.

### 2.9. Identified Potential Sources of Horizontal Transfer Candidates

Using the same “search, extend, align, trim” method described above we curated consensus sequences of the HTT candidate sequences from all genomes containing hits of 70% or higher pairwise identity to, and 50% coverage of, the consensus sequences as found. By lowering the threshold to 70% pairwise identity we aimed to identify divergent repeats to be used as outgroups. In some species no consensus sequence could be constructed due to very low copy number and/or very high degradation of the repeats identified. Additionally, when viewing some MSAs it became clear two distinct subfamilies were present. In these cases the MSAs were split into separate MSAs for each subfamily and separate consensuses were made for each.

To create phylogenies for each HTT candidate we constructed phylogenies using FastTree 2.1.1.1 [41] from nucleotide MSAs of the consensus sequences aligned using MAFFT v7.453 [32] (localpair). The phylogenies were viewed for analysis using FigTree v1.4.4 [42].

## 3. Results and Discussion

### 3.1. Draft Terrestrial Elapid Genomes Are Fragmented but Show High Completeness

We generated primary 10× linked-read assemblies of the two terrestrial elapids of good quality in terms of scaffold sizes and completeness estimates (Table 1). The eastern brown snake (*P. textilis*) genome assembled into a 1.59 Gb haploid reference on 28,550 primary scaffolds (2.47% gaps), with over 50% of the genome on the 31 longest scaffolds (N50 = 14.69 Mb) and a BUSCO v5 MetaEuk completeness score (sauropsida_odb10, *n* = 7840) of 91.0% (1.7% duplicated, 2.1% fragmented, 6.9% missing). The 504 heterozygous alternative scaffolds covered 1.23 Gb (77.7%) of the assembly. The mainland tiger snake (*N. scutatus*) genome assembled into a 1.67 Gb haploid reference on 52,414 primary scaffolds (4.84% gaps), with over 50% of the genome on the 66 longest scaffolds (N50 = 5.997 Mb) and a BUSCO v5 MetaEuk completeness score (sauropsida_odb10, *n* = 7840) of 87.9% (1.6% duplicated, 3.3% fragmented, 8.8% missing). The 1158 heterozygous alternative scaffolds covered 1.26 Gb (75.7%) of the assembly. The assembly quality of both snake genomes compares well with the other snake genomes used in the study [43,44,45,46]. The quality of both snake assemblies was sufficient for annotation and inclusion in NCBI RefSeq and Ensembl (Table 1).

### 3.2. Genome Quality Affects Repeat Annotation

Previous studies have highlighted the importance of genome assembly quality in repeat annotation, with higher sequencing depths and long-read technologies critical for resolving TEs [47,48]. Our repeat analysis reveals significant variation in total TE content between genome assemblies (Table 2, Figure 1, Appendix A), however, some of this variation is likely due to large differences in assembly quality rather than differential TE expansions or contractions in certain lineages. Most notably, the TE content of the *Ophiophagus* assembly is significantly lower than that of the other species (~36% compared with ~46%). The TE content of the *Aipysurus* assembly is also notably lower, however to a lesser extent (41% compared with ~46%). The *Naja*, *Laticauda*, *Notechis*, and *Pseudonaja* assemblies are much higher quality assemblies than the *Ophiophagus* and *Aipysurus* assemblies, having longer contigs and scaffolds (Table 1). This discrepancy is because the *Ophiophagus* and *Aipysurus* genomes are both assembled solely from short-read data with a low sequencing depth (28× and 30×, respectively). In stark contrast, the *Naja* genome is assembled from a combination of long-read (PacBio and Oxford Nanopore) and short read (Illumina) data, scaffolded using Chicago and further improved using Hi-C and optical mapping (Bionano) technologies. In the middle ground, the *Laticauda*, *Notechis* and *Pseudonaja* assemblies utilize a combination of 10× Chromium linked-read and short-read technologies. Many of the recently expanded TEs in the *Ophiophagus* and *Aipysurus* genomes likely collapsed during assembly because of their very high sequence similarity. Therefore, the apparent lack of recent activity in *Ophiophagus* and *Aipysurus* is a likely artefact of assembly quality. As the total TE content annotated in the *Naja*, *Laticauda*, *Notechis* and *Pseudonaja* is comparable at 46-48% of the genome and the four genomes are of comparable quality, the majority of the following analyses focuses on these four species.

### 3.3. Recent Insertions vs. Ancestral Insertions

Recent TE insertions are likely to have diverged only slightly from the sequences Repeat Masker used to identify them, while ancestral insertions are likely to be highly divergent. Based on this assumption, we attempted to differentiate between recent and ancestral insertions using the pairwise identity of TE insertions to the mc-subfamily used to identify them. To estimate the expected divergence of ancestral TE insertions from consensus sequences compared to new insertions we searched for orthologues of 5000 randomly selected *Notechis* TE insertions and their flanks in the *Naja* assembly (Figure 2). From the 5000 TEs we were able to identify 2192 orthologues in *Naja*. As expected, the median pairwise percent identity of the ancestral TEs to the curated consensus sequences was notably lower compared with that of all TEs, at 90.5% compared to 93.6%. Similarly, the 95% quantile of pairwise percent identity was also lower (for ancestral TEs compared to all TEs), at 96.2% compared to 98.4%. Based on these results we treated TEs with a percent identity of 96% or higher to consensus sequences as having likely been inserted since divergence from Asian elapids (i.e., the TEs appear too similar to the mc-subfamily used to identify them to represent ancestral insertions).

### 3.4. Recent Expansion of Specific Superfamilies

By comparing TE divergence profiles of the various assemblies, we can gain an overall picture of how TE superfamilies have expanded since the split of Hydrophiinae from Asian elapids (Figure A1, Figure A2 and Figure A3 in Appendix B). Examining LTR retrotransposons, we see that large expansions of *Gypsy* elements are apparent in both the *Naja* and hydrophiine assemblies, however, *Copia* and DIRS elements and ERVs appear inactive in *Naja* while expanding in hydrophiines. The divergence profile of DNA transposons suggests *Tc1-Mariner* and *hAT* transposons to have either been expanding at a similar rate in all species and/or result from ancestral expansion, with the exception of the explosive expansion of *PIF-Harbinger* transposons in *Laticauda* (see [7]). The greatest variation was seen within non-LTR retrotransposons, with L2s highly active in *Naja* yet completely inactive in hydrophiines. Instead, multiple other LINE superfamilies expanded in hydrophines, in particular CR1s and L1s. While similar differences in TE expansions between snake lineages have been reported by Castoe et al. [22], Yin et al. [51] and Pasquesi et al. [21], the expansions we describe here are over much shorter time periods.

Without highly contiguous assemblies of all species it is difficult to comprehensively identify and quantify recent or ongoing TE expansions. However, we propose to use transcription as a proxy for transposition where we identify currently expressed TE families in present day species as being active and potentially expanding. To test this hypothesis, we first identified TE subfamilies in each species with over 100 kb of copies with >95% pairwise identity to the consensus sequences used to identify them; treating these subfamilies as potentially expanding. By mapping raw transcriptome reads back to these consensuses, we were able to identify expressed TE subfamilies. In all four species, diverse TEs are currently expressed including subfamilies of *Copia*, ERV, DIRS, *Gypsy*, Penelope, CR1, L1, *Rex1*, RTE, *hAT* and *Tc1-Mariner*.

### 3.5. Continued Expansion or Horizontal Transfer

The TE subfamilies which we have identified as recently expanded within Hydrophiinae could be ancestral and continuously expanding since diverging from Asian elapids or have been horizontally transferred from long diverged species. Differentiating between ancestral and horizontally transferred TEs is difficult, and the supporting evidence must meet strict conditions [52]. Horizontally transferred sequences are defined as having a patchy phylogenetic distribution, a higher level of similarity to sequences in another species than would be expected based on divergence time and a sequence phylogeny incompatible with vertical inheritance. To identify any TEs which may have been horizontally transferred into Hydrophiinae, we conservatively estimated the expected minimum pairwise identity of TEs present in both hydrophiines and Asian elapids using the 2192 orthologous sequences identified in *Notechis* and *Naja* to be 80% (Figure 3). Based on this pairwise identity, any vertically inherited TE subfamily classified as recently expanding in hydrophiines will likely have copies of 80% pairwise identity present in Asian elapids.

To determine whether any recently expanding TE subfamilies were horizontally transferred into hydrophiines following their divergence from Asian elapids, we searched for them in the genomes of *Naja*, *Ophiophagus* and an additional eight non-elapid snakes. Some recently expanding subfamilies absent from *Naja* and *Ophiophagus* were present in non-elapid snakes at 80% or higher identity. To be conservative we treated these TEs as ancestral, likely being lost from Asian elapids. The remaining TE subfamilies, those present in hydrophiines but absent from other snakes, were treated as horizontal transfer candidates. To confirm these candidate TEs were horizontally transferred into hydrophiines we searched for them in the over 2300 GenBank metazoan reference and representative genome assemblies with scaffold N50s > 100 kb available (see Appendix A for assembly accessions, Appendix A for BLAST output). This search revealed thirteen autonomous TEs present in non-serpentine genomes at 80% or higher identity that are therefore likely to have been horizontally transferred into hydrophiines. In addition, curation revealed an additional HTT candidate similar to a *hAT* initially identified in *Notechis* to be present in hydrophiines which had not been identified by RepeatModeler.

Of these fourteen HTT candidates, four were transferred into the ancestral hydrophiine, seven into sea kraits since their divergence from other hydrophiines, one into the common ancestor of terrestrial hydrophiines and sea snakes, one into sea snakes and one into *Notechis* following its divergence from sea snakes (Figure 4A). To determine potential sources of these repeats we, where possible, manually curated consensus sequences of the similar repeats identified in the other metazoan genomes using the search-extend-align-trim methodology described above. A lower threshold was used to identify divergent repeats to be used as outgroups. Additionally, any divergent repeats could help clarify if the repeats were present in the ancestors of the potential source species, or if they were horizontally transferred into these species as well. For each HTT candidate subfamily, the nucleotide sequences of curated repeats were aligned (Appendix A), and the alignments used to construct phylogenies (Appendix A). From these phylogenetic trees we were able to establish potential sources of the HTT candidates.

We have previously described two of the fourteen HTT events in detail, that of *Proto2-Snek* to *Aipysurus* and *Harbinger-Snek* to *Laticauda*, both of which were likely transferred from a marine species (see [6,7]). The six newly identified HTT events in *Laticauda* were probably also transferred from an aquatic species, as similar sequences are found only in marine and freshwater species (fish, tunicates, frogs and a freshwater turtle).

Particularly interesting is a *hAT* unique to *Notechis*, suggesting it was transferred following its divergence from sea snakes (Figure 4B). A *hAT* identified in the eastern banjo frog (*Limnodynastes dumerilii*) shows over 99% pairwise identity to the *hAT* unique to *Notechis*. Less similar copies (95–97%) were found in multiple species of Eurasian frogs. The very high similarity of the *hAT* found in *Notechis* and *Limnodynastes,* with *Limnodynastes* being a major prey items of *Notechis* [53] suggests the very recent horizontal transfer of the *hAT* between *Notechis* and *Limnodynastes*, however the direction of transfer has not been confirmed. Alternatively, this transfer could be two independent transfers from a shared pathogen or parasite.

The *Rex1* horizontally transferred into the common ancestor of terrestrial hydrophiines and sea snakes was only identified elsewhere in the central bearded dragon (*Pogona vitticeps*), an agamid lizard native to the inland woodlands and shrublands of eastern and central Australia [54]. As this TE appears restricted to Australian squamates, this HTT event is likely to have occurred after hydrophiines reached Australia and before the transition of sea snakes to their marine habitat.

Three of the four repeats common to all hydrophiines (the *hAT*, *Gypsy* and *Rex1*) appear to have been horizontally transferred in an aquatic environment; with the most similar retrotransposons being identified in a damselfish, eel and a catfish, respectively (Figure 5). As both eels and catfish are prey of extant sea kraits and sea snakes [55,56,57], these retrotransposons could have been transferred into the ancestral hydrophiinae from related eels or catfish. We can use this new understanding of elapid TE evolution to understand hydrophiine evolution and adaptation to their marine environment. Marine elapids (sea kraits and sea snakes) and terrestrial Australian elapids were originally considered two distinct lineages [58,59,60], however, the recent adoption of molecular phylogenomics has resolved Hydrophiinae as a single lineage, with sea kraits as a deep-branch and sea snakes nested within terrestrial Australian snakes [1,61,62]. Fossil evidence combined with an understanding of plate tectonics has revealed Hydrophiinae, like many other lineages of Australian reptiles, likely reached Australia via islands formed in the Late Oligocene-Early Miocene by the collision of the Australian and Eurasian plates [63,64,65,66,67]. Alternatively, it has also been proposed the common ancestor of Hydrophiinae may have been a semi-marine “proto-*Laticauda*”, which reached Australia in the Late Oligocene directly from Asia [68]. The horizontal transfer of these three TEs into the ancestral hydrophiine, likely from a marine organism, provides tangible support for the hypothesis that the ancestral hydrophiine was a semi-marine or marine snake.

## 4. Conclusions

In our survey of elapid genomes, we have found that TE diversity and their level of expansion varies significantly within a single family of squamates, similar to the variation previously seen across all squamates or within long diverged snakes. This diversity and variation is much greater than what has been reported for mammals and birds. Our finding of HTT into lineages of Hydrophiinae exposed to marine habitats’ environments, indicates that novel environments may play a large role in HTT through exposure to new TEs. Additionally, the HTT of three TEs found solely in marine organisms into the ancestral hydrophiine provides the first evidence that terrestrial Australian elapids are likely derived from a marine or amphibious ancestor. 

As long-read genome sequencing becomes feasible and cost-effective for more species, genome assembly quality will continue to increase and the sequencing of multiple genomes of non-model organisms will become commonplace. Using these higher quality genomes as they become available, will allow a better understanding of HTT and the role that TEs play in adaptive evolution. We highlight how hydrophiinae provide an ideal system for such studies due to their rapid adaptation to a wide range of environments and the multiple HTT events we have identified.

## Figures and Tables

**Figure 1 genes-13-00217-f001:**
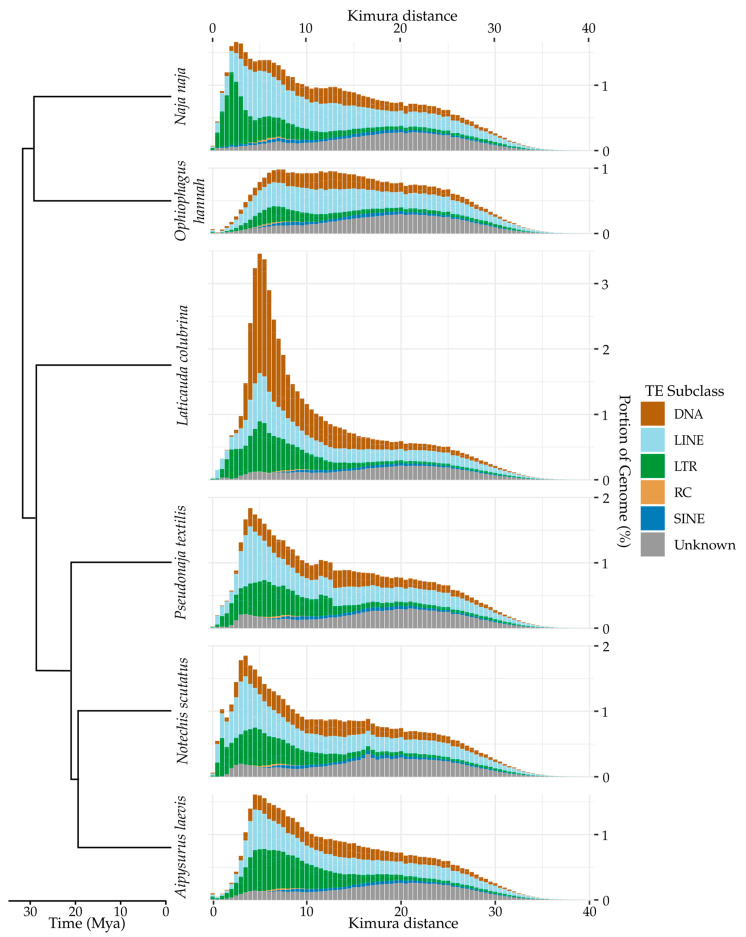
Overall TE divergence profile of four hydrophiines and two Asian elapids. Ancestral TE expansion is similar across hydrophiines and Asian elapids while recent expansion varies between species. Due to much lower genome assembly quality resulting in collapsed TEs, little recent expansion in the *Aipysurus laevis* and *Ophiophagus hannah* genomes was detected. TEs were identified using Repeat Masker [49] and a custom repeat library (see methods).

**Figure 2 genes-13-00217-f002:**
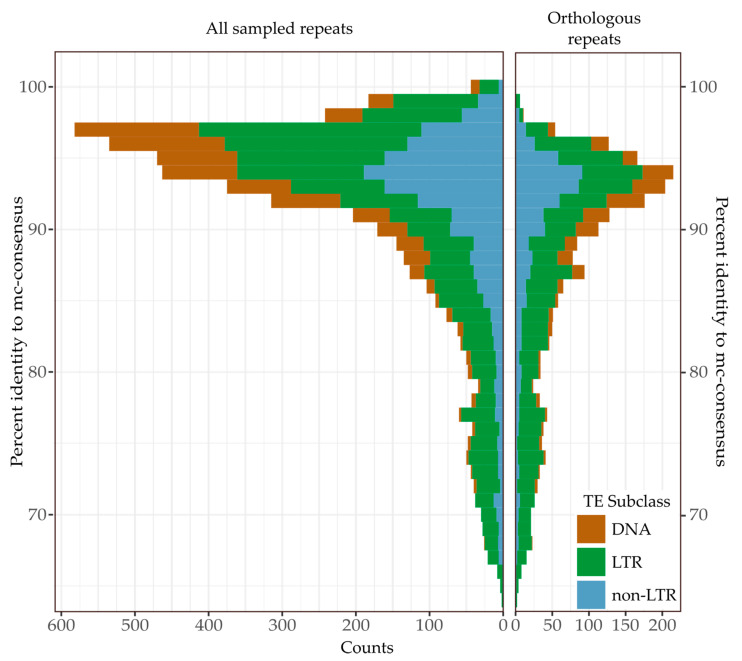
Similarity to consensus of 5000 randomly selected TE insertions in *Notechis scutatus* compared to that of the subset of 2192 TE insertions having orthologues in *Naja naja*. The similarity of ancestral insertions from mc-subfamily consensuses used to identify them was notably lower than that of TEs likely inserted since the species diverged. TEs were initially identified in *Notechis* using Repeat Masker [49]. The presence of orthologues in *Naja* was determined using BLASTN (-task dc-megablast) [31].

**Figure 3 genes-13-00217-f003:**
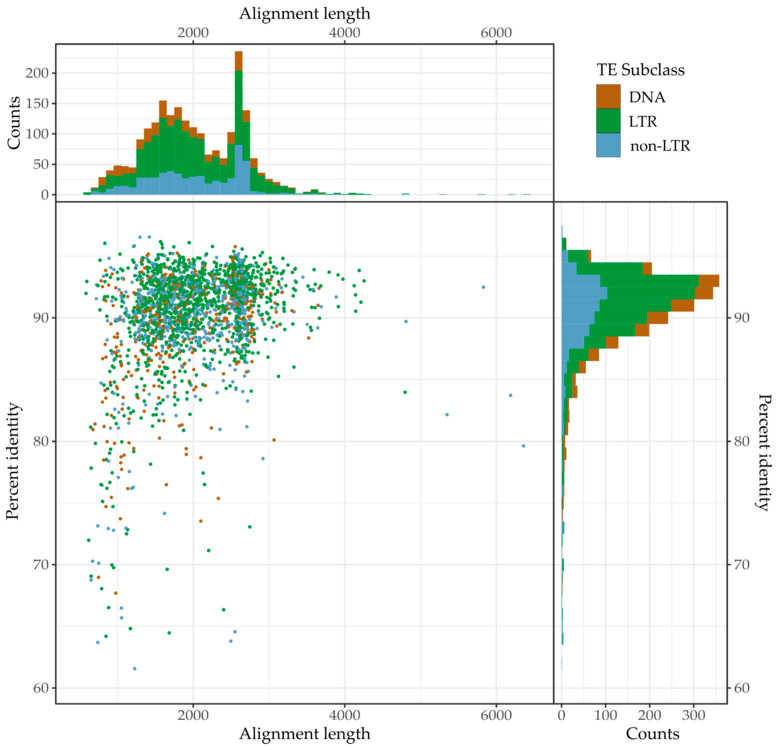
**Similarity (% pairwise identity) of orthologous TEs in *Notechis scutatus* and *Naja naja* genomes.** TEs were initially identified in *Notechis scutatus* using Repeat Masker [49], with orthologues identified in and pairwise identity calculated for *Naja naja* using BLASTN (-task dc-megablast) [31].

**Figure 4 genes-13-00217-f004:**
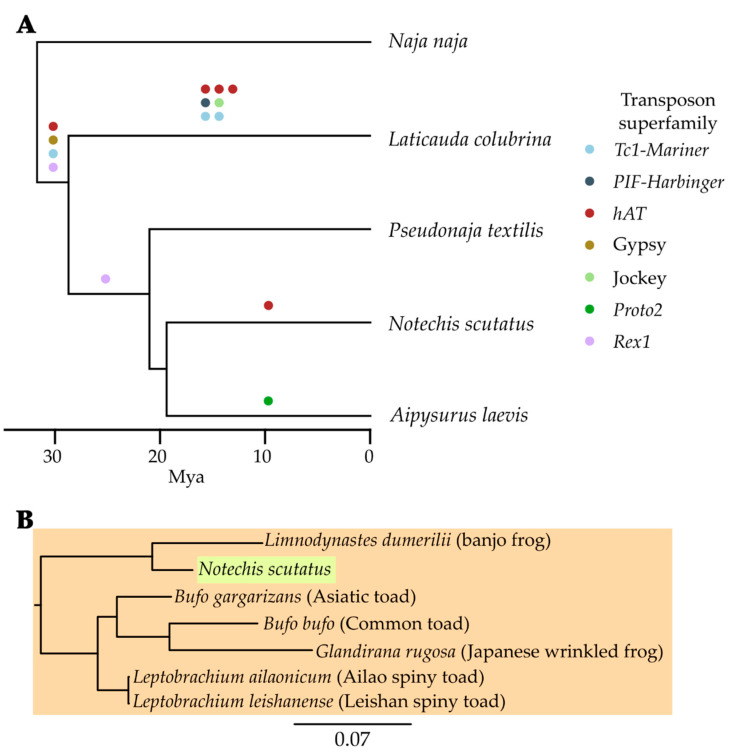
**Horizontal transfer of TEs into hydrophiines since their divergence from Asian elapids.** (**A**) At least fourteen autonomous TEs have been horizontally transferred into hydrophiines, most likely from marine organisms. We previously described the horizontal transfer of the *Proto2* to *Aipysurus* in [6] and the *PIF-Harbinger* to *Laticauda* in [7]. (**B**) An example of recently transferred *hAT* transposon identified in *Notechis scutatus* but absent from other hydrophiines. Transposons identified in amphibians are highlighted in orange, and that in the snake in green. This transposon was potentially transferred from a prey item, the eastern banjo frog (*Limnodynastes dumerilii*). BLASTN [31] searches of the *Limnodynastes* genome identified sequences showing 99.7% identity to those found in *Notechis*. For the full trees of similar TEs identified see Data S4.

**Figure 5 genes-13-00217-f005:**
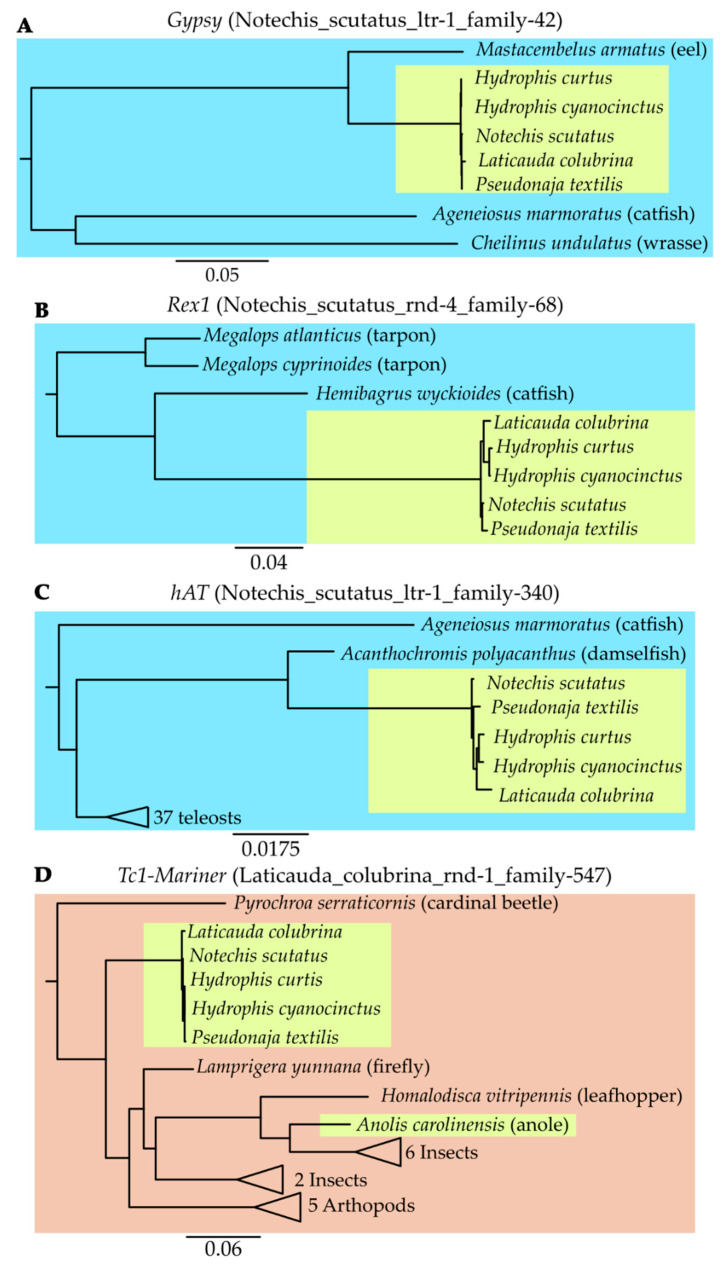
**Extract from the phylogenies of the TEs horizontally transferred into the ancestral hydrophiine and the most similar sequences identified in over 2300 metazoans.** (**A**–**C**) The absence of similar TEs from other elapids combined with high similarity to TEs identified in fish suggests each HTT event occurred from a fish into the ancestral hydrophiine following their divergence from Asian elapids. Transposons identified in marine species are highlighted in blue, and that in the hydrophiines in green. (**D**) Similarly, the absence of similar TEs from other elapids combined with high similarity to TEs identified in arthropods suggests the HTT event occurred from an arthropod into the ancestral hydrophiine following their divergence from Asian elapids. Transposons identified in arthropods are highlighted in orange, and that in hydrophiines and a lizard in green. For full trees see Data S4.

**Table 1 genes-13-00217-t001:** Assembly statistics for the snake genomes used in this study.

	*Naja naja*	*Ophiophagus hannah*	*Laticauda colubrina*	*Pseudonaja textilis*	*Notechis scutatus*	*Aipysurus laevis*
No. of scaffolds	1897	296,399	20,583	28,550	52,414	196,089
Assembly size	1.77 Gb	1.59 Gb	2.04 Gb	1.59 Gb	1.67 Gb	1.85 Gb
Min. scaffold	585 bp	200 bp	1000 bp	1000 bp	1000 bp	179 bp
Max. scaffold	375.0 Mb	2.845 Mb	114.7 Mb	55.11 Mb	50.92 Mb	529.6 kb
Scaffold N50	224.1 Mb	241.5 kb	39.96 Mb	14.69 Mb	5.997 Mb	24.81 kb
Scaffold L50	3	1750	16	31	66	22,027
No. contigs	13,805	816,633	105,298	88,022	131,892	310,523
Contig N50	277.2 kb	4.170 kb	34.05 kb	48.75 kb	29.62 kb	10.89 kb
Contig L50	1587	94,154	16,388	8,636	14,776	43,489
% Gaps	6.20%	13.4%	8.88%	2.47%	4.84%	13.6%
% GC	40.46%	39.46%	41.25%	39.84%	39.72%	39.90%
BUSCO v5 Completeness (sauropsida_odb10, *n* = 7840)	90.0%	84.5%	90.9%	91.0%	87.9%	59.9%
- Single-copy	88.6%	83.6%	89.9%	89.3%	86.3%	58.9%
- Duplicated	1.4%	0.9%	1.0%	1.7%	1.6%	1.0%
- Fragmented	1.9%	5.5%	2.3%	2.1%	3.3%	18.7%
- Missing	8.1%	10.0%	6.8%	6.9%	8.8%	21.4%
GenBank Accession	GCA_009733165.1	GCA_000516915.1	GCA_015471245.1	GCA_900518735.1	GCA_900518725.1	-
Ensembl Identifier	Nana_v5	-	-	EBS10Xv2-PRI	TS10Xv2-PRI	-
First published	Suryamohan et al. [43]	Vonk et al. [44]	Kishida et al. [45]	This study	This study	Ludington and Sanders [46]

**Table 2 genes-13-00217-t002:** Interspersed repeat composition (% of genome) of hydrophiine and Asian elapid genome assemblies. Variation in assembly repeat content varies both within hydrophiines and between hydrophiines and Asian elapids. Genome assemblies were annotated with Repeat Masker [49] and a custom library of curated RepeatModeler2 libraries [30] and previously described lepidosaur TEs from the Repbase Repeat Masker library [50].

	*Naja naja*	*Ophiophagus hannah*	*Laticauda colubrina*	*Pseudonaja textilis*	*Notechis scutatus*	*Aipysurus laevis*
**Class I (Retrotransposons)**	30.58	19.73	23.00	27.81	27.29	25.91
Penelope	1.72	1.37	2.08	2.06	1.98	1.64
LINE/CR1	4.37	4.12	4.12	5.54	6.28	4.67
LINE/L1	3.24	2.21	2.94	4.04	3.40	2.72
LINE/L2	7.14	3.39	1.05	1.41	1.4	1.27
LINE/Rex-Babar	1.17	1.11	1.08	1.44	1.42	1.26
LINE/RTE	1.40	1.52	1.07	1.50	1.43	1.25
LINE/Other	0.62	0.69	0.54	0.66	0.64	0.59
SINE	0.37	0.43	0.30	0.40	0.39	0.36
LTR/Copia	0.71	0.47	1.02	1.27	1.65	0.89
LTR/DIRS	0.81	0.61	0.86	1.04	1.19	1.57
LTR/ERV	0.60	0.42	1.23	1.38	1.69	1.16
LTR/Gypsy	7.01	1.97	5.6	5.61	4.38	7.22
LTR/Other	1.42	1.42	1.11	1.46	1.44	1.31
**Class II (DNA Transposons)**	7.81	7.45	18.08	9.02	9.20	7.95
DNA/hAT	4.25	4.13	3.78	5.09	5.13	4.47
DNA/PIF-Harbinger	0.02	0.03	11.19	0.02	0.02	0.02
DNA/Tc1-Mariner	3.13	2.88	2.79	3.47	3.63	3.08
DNA/Other	0.22	0.24	0.17	0.24	0.22	0.20
RC/Helitron	0.19	0.17	0.15	0.20	0.20	0.18
**Unknown**	7.93	8.57	6.46	9.18	9.26	7.89
**Total interspersed repeats**	46.32	35.75	47.54	46.01	45.75	41.75

## Data Availability

The data presented in this study are freely available in Zenodo at https://doi.org/10.5281/zenodo.5820601 (accessed on 6 January 2022).

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
