# Peer review of "Horizontal Transposon Transfer and Its Implications for the Ancestral Ecology of Hydrophiine Snakes"

_genes, 2022, doi:10.3390/genes13020217_

Round 1
Reviewer 1 Report
This is a high quality manuscript presenting still very novel and above all interesting results on horizontal transfer of transposons (HTT) in a marine lineage of snakes.
Authors first sequenced and assembled relevant genomes. Authors described their elaborated computational methods in much details and did not forget to mention that the genome assembly quality is crucial and a limiting factor to their analyses. They analysed their data accordingly and also aware of the necessary prerequisites to be eligible to assume HTT in the selected elements.
The manuscript is well prepared and the images are of high quality, hence, there is nothing to require to be improved.
However, it is such an interesting and at the same time novel topic that an outstanding question remains throughout the entire manuscript - how should the reader imagine the trajectory of the TE from the digestive apparatus of the snakes (as the TEs probably originate from the prey, right?) towards germ line cells (so that they would become incorporated in the genome also for the following generations)? Is there any hypothesis or scenario proposed? This is missing in the manuscript and should be explained to make the manuscript complete.
Author Response
This has been a question in HTT research for many years and seeing as we don’t examine any mechanics of HTT here we are not comfortable with speculation. This sort of question/speculation would be more suited for a review of HTT or research papers directly investigating this, say, with cell cultures, rather than papers finding the occurrence of HTT.
We do indicate that the transfers in the frog could have gone either way, or “alternatively this transfer could be two independent transfers from a shared pathogen or parasite”. In regards to the others the pairwise identity/genetic distance is lower, so while the transfers are likely from marine organisms into snakes we can’t be certain with respect to vector or mechanism (prey, pathogen or parasite).
Reviewer 2 Report
In this study the authors have used genome sequences from six snakes to investigate the origin and the distribution of TEs in the genomes. I find this work exploratory as the assembly quality highly affects the accurate detection of the TEs in the genomes. Despite that, they have performed a thorough analysis on the dynamics of various TE families in the genomes. They provide evidence that some TEs have been horizontally transferred from other animals with which the particular species are connected (basically as predator-prey relationship). I find the research well conducted, the article concise and well written and therefore I consent to its publication
A minor comment: I cannot find the figures A1-A3 mentioned in Line 306.
Author Response
Figures A1 to A3 embedded in line in Appendix A, before the Supplementary Material section of the manuscript document.
Reviewer 3 Report
The authors Galbraith et al. presented a manuscript in which they attend transposable element diversity and expansion in hydrophiines and Asian elepsids and identify multiple horizontal transfer events in these species, using bioinformatic analyses.
I consider that the scientific topic being addressed as interesting, methodology appropriate .
My comments and suggestions for the authors are listed below:
Maybe the authors can find the publication of Ahmad et al. (Cells 2021; 10,7,1707) of use, in the context of TE content and abundance in snakes in the Introduction (Line 98) and/or the Results and Discussion section.
Line 175: It is not explained why Notechis scutatus and Naja naja genomes were chosen for this analysis. For Naja naja one could suppose assembly quality was a criteria, however, what was the rationale for choosing Notechis scutatus from the other group?
Line 175 and further throughout the text: The authors use the name of the genus (Notechis and Naja) while speaking about genome assemblies of specific species, Notechis scutatus and Naja naja. Please correct throughout the manuscript the usage of genus name when referring to a specific species.
Line 201-202: What are the names of the species inspected? In the Results section “eight non-elapid snakes” are mentioned.
Line 269-270: It would be more appropriate to place the sentence “Variation in assembly repeat content varies both within hydrophiines and between hydrophiines and Asian elapids” in the main text rather than in the Table title.
Lines 292-294: The authors state that little recent expansion in the Aipysurus laevis and Ophiophagus hannah genomes was detected due to their much lower genome assembly quality which collapsed TEs of high sequence similarity. However, in Figure 1 it seems that the number of elements having Kimura distance 0 is larger for Aipysurus laevis than for Laticauda colubrina, Pseudonaja textilis and Notechis scutatus.
In case TEs of high sequence similarity are considered to be collapsed in Aipysurus laevis, shouldn't this species be excluded from the following analyses as, when compared to the data from other species, wrong conclusions could be brought:
1) the analysis of recent expansion of specific superfamilies (Line 303 and Figures A1-A3),
2) currently expressed TE
3) HTT analysis
I suggest the section of Results and Discussion regarding recent expansion of specific superfamilies (Lines 304-315) and results presented in Figure A1-3 to be discussed in more detail.
Two peaks noticeable at Figure A1 for Tc1/Mariner elements could be addressed, one recent cycle of expansion resulting in a peak with Kimura distance <5 and one previous cycle, now with accumulated mutations and Kimura distance between 10 and 15. hAT elements seem to have a different history of activity compared to Tc1/Mariner. Also, CR1 elements in Notechis scutatus have an interesting profile, differing from other species.
The information regarding materials and methods used are not necessary in Figure descriptions, e.g. lines: 294, 301-302, 492, etc.
In the Materials and methods section the authors declare the source of Notechis scutatus and Pseudonaja textilis. Although not explicitly mentioned in the Materials and methods section the data for other species is presumably retrieved from publicly available databases, as their GenBank accesion numbers are brought in Table 1 of the Results. Is the missing Aipysurus laevis GenBank Accession number PRJNA669384?
“Figure 4” is missing from Figure 4 description.
Both HTT and HT abbreviations are used in the text when referring to the horizontal (transposon) transfer, please unify and use only one abbreviation throughout the manuscript.
Author Response
Reviewer 3 Comments and Suggestions for Authors. Our responses are in BLUE
The authors Galbraith et al. presented a manuscript in which they attend transposable element diversity and expansion in hydrophiines and Asian elepsids and identify multiple horizontal transfer events in these species, using bioinformatic analyses.
I consider that the scientific topic being addressed as interesting, methodology appropriate .
My comments and suggestions for the authors are listed below:
Maybe the authors can find the publication of Ahmad et al. (Cells 2021; 10,7,1707) of use, in the context of TE content and abundance in snakes in the Introduction (Line 98) and/or the Results and Discussion section.
While this review paper is relevant to TE evolution in snakes, we have cited the same original research that review did, largely our own and from Castoe’s lab.
Line 175: It is not explained why Notechis scutatus and Naja naja genomes were chosen for this analysis. For Naja naja one could suppose assembly quality was a criteria, however, what was the rationale for choosing Notechis scutatus from the other group?
We were examining how the TE landscape of hydrophiines (which include all sea kraits, sea snakes and Australian elapids) have changed since their divergence from Asian elapids. Notechis scutatus was included as it is an Australian elapid, hence a hydrophiine. Naja naja is an outgroup with a high quality assembly. If the reviewer is wondering why two cobras were used it’s because they’re both Asian elapids.
Line 175 and further throughout the text: The authors use the name of the genus (Notechis and Naja) while speaking about genome assemblies of specific species, Notechis scutatus and Naja naja. Please correct throughout the manuscript the usage of genus name when referring to a specific species.
In the interests of returning the revised manuscript as quickly as possible, we have opted to indicate in the introduction what the use of genus name means
“ (henceforth all mentions of these species will use the genus name only, ie Aipysurus for A. laevis)”
Line 201-202: What are the names of the species inspected? In the Results section “eight non-elapid snakes” are mentioned.
Table S1 includes the assembly names and accession numbers of all genomes used, including these eight non-elapids.
Line 269-270: It would be more appropriate to place the sentence “Variation in assembly repeat content varies both within hydrophiines and between hydrophiines and Asian elapids” in the main text rather than in the Table title.
This is the table title and this point is covered in the text.
Lines 292-294: The authors state that little recent expansion in the Aipysurus laevis and Ophiophagus hannah genomes was detected due to their much lower genome assembly quality which collapsed TEs of high sequence similarity. However, in Figure 1 it seems that the number of elements having Kimura distance 0 is larger for Aipysurus laevis than for Laticauda colubrina, Pseudonaja textilis and Notechis scutatus.
In case TEs of high sequence similarity are considered to be collapsed in Aipysurus laevis, shouldn't this species be excluded from the following analyses as, when compared to the data from other species, wrong conclusions could be brought:
1) the analysis of recent expansion of specific superfamilies (Line 303 and Figures A1-A3),
2) currently expressed TE
3) HTT analysis
We are a little confused by this comment. If the assembly says more 0% distance copies are present (assembled), why would we need to exclude the species? Consider that under assembly of 0% distance TES will always happen, even in high coverage, long range assemblies.
Therefore we should not exclude the olive sea snake. We have mentioned potential issues that might affect our analysis in the text and our methods are conservative in order to take these into account so our conclusions are sound. In particular, the transcriptome sequences are a distinct data set, so assembly quality is not relevant to the presence of TE mRNA in transcriptomes. Finally, if BLAST searches find that a TE is present in other Australian elapids fub not in Aipysurus it means that it is either not assembled or has been lost, and does not impact the HTT analysis.
I suggest the section of Results and Discussion regarding recent expansion of specific superfamilies (Lines 304-315) and results presented in Figure A1-3 to be discussed in more detail.
Two peaks noticeable at Figure A1 for Tc1/Mariner elements could be addressed, one recent cycle of expansion resulting in a peak with Kimura distance <5 and one previous cycle, now with accumulated mutations and Kimura distance between 10 and 15. hAT elements seem to have a different history of activity compared to Tc1/Mariner. Also, CR1 elements in Notechis scutatus have an interesting profile, differing from other species.
With respect, we don’t think this is needed. The main point in A1 is comparison of Pif vs Tc1 showing post HTT expansion. For the other “differences”, we think readers should make up their own minds when looking at the figures. Furthermore, if two peaks are present for one subfamily, it is possible that it would need to be split into two subfamilies. For the purposes of our analysis, this is not required.
The information regarding materials and methods used are not necessary in Figure descriptions, e.g. lines: 294, 301-302, 492, etc.
This information is important for readers who have not read the methods section or who are skimming the paper. The journal guidelines for manuscript submission indicate that methods should be mentioned in the figure captions so we are following those guidelines.
In the Materials and methods section the authors declare the source of Notechis scutatus and Pseudonaja textilis. Although not explicitly mentioned in the Materials and methods section the data for other species is presumably retrieved from publicly available databases, as their GenBank accesion numbers are brought in Table 1 of the Results. Is the missing Aipysurus laevis GenBank Accession number PRJNA669384?
The “missing” data is not available from GenBank BioProject PRJNA669384, but is instead available from the SI of the relevant paper referenced in Table 1.
“Figure 4” is missing from Figure 4 description.
In our version of the manuscript document, Figure 4 clearly says Figure 4.
Both HTT and HT abbreviations are used in the text when referring to the horizontal (transposon) transfer, please unify and use only one abbreviation throughout the manuscript.
We have changed HT to HTT for all mentions.